# Sina Score as a New Machine Learning-Derived Online Prediction Model of Mortality for Cirrhotic Patients Awaiting Liver Transplantation: A Prospective Cohort Study

**DOI:** 10.3390/jcm14134559

**Published:** 2025-06-27

**Authors:** Seyed Mohammad Kazem Hosseini-Asl, Seyed Jalil Masoumi, Ghazaleh Rashidizadeh, Amir Hossein Hassani, Golnoush Mehrabani, Vahid Ebrahimi, Seyed Ali Malek-Hosseini, Saman Nikeghbalian, Alireza Shakibafard

**Affiliations:** 1Department of Internal Medicine, School of Medicine, Shiraz University of Medical Sciences, Shiraz 7193613311, Iran; hoseiniasl@sums.ac.ir; 2Nutrition Research Center, Department of Clinical Nutrition, School of Nutrition and Food Sciences, Shiraz University of Medical Science, Shiraz 7134814336, Iran; 3Gastroenterohepatology Research Center, Shiraz University of Medical Sciences, Shiraz 7134814336, Iran; 4Center for Cohort Study of SUMS Employees’ Health, Shiraz University of Medical Sciences, Shiraz 7134814734, Iran; 5Student Research Committee, School of Nutrition and Food Sciences, Shiraz University of Medical Sciences, Shiraz 7153675500, Iran; ghazale.rz@gmail.com; 6School of Medicine, Shiraz University of Medical Sciences, Shiraz 7134845794, Iran; amirhossein.hassani77@gmail.com; 7Department of Rehabilitation Medicine, University of Alberta, Edmonton, AB T6G 2G4, Canada; 8Department of Biostatistics, School of Medicine, Shiraz University of Medical Sciences, Shiraz 7134845794, Iran; biostat.ebrahimiv@gmail.com; 9Shiraz Transplant Research Center, Shiraz University of Medical Sciences, Shiraz 7193635899, Iran; malekha@sums.ac.ir (S.A.M.-H.); nikeghbals@gmail.com (S.N.); 10TABA Medical Imaging Center, Shiraz 7134846114, Iran; drshakibafard@gmail.com

**Keywords:** mortality prediction, liver cirrhosis, machine learning approach, Sina score, MELD

## Abstract

**Background:** Cirrhosis is responsible for a large proportion of mortality worldwide. Despite having multiple scoring systems, organ allocation for end-stage liver disease remains a major problem. Since anthropometric indices play important roles in predicting the prognosis of patients with cirrhosis, these variables were used in establishment of a novel scoring system. **Methods:** In order to evaluate a machine learning approach for predicting the probability of three-month mortality in cirrhotic patients awaiting liver transplantation, the clinical and anthropometric information of 64 patients referred to Abu-Ali-Sina Transplantation Center were collected and followed for three months. A LASSO logistic regression model was used to devise and validate a new machine learning approach and compare it to the Model for End-Stage Liver Disease (MELD) regarding the three-month mortality of cirrhotic patients. Hand grip, skeletal muscle mass index, average mean arterial pressure, serum sodium, and total bilirubin were assessed with this new machine learning approach to predict the prognosis of patients with cirrhosis, which we named the Sina score. **Results:** Sixty-four patients were enrolled, with a mean age of 46.50 ± 12.871 years. Like the MELD score, the Sina score is a precise prognostic tool for predicting the three-month mortality probability in cirrhotic patients [area under the curve (AUC) = 0.753 and *p* = 0.005 vs. AUC = 0.607 and *p* = 0.238]. Our machine learning approach, the Sina score, was shown to be a precise prognostic tool, like the MELD, for the prediction of the three-month mortality probability of cirrhotic patients awaiting liver transplantation. **Conclusions:** The Sina score, given that its level of precision is on par with the MELD, can be recommended for the prediction of three-month mortality in cirrhotic patients awaiting liver transplantation.

## 1. Introduction

Cirrhosis, or end-stage liver disease, affected more than 120 million people worldwide in 2017, while approximately 10 million suffered from decompensated cirrhosis [1]. The prevalence and incidence of cirrhosis is trending upwards due to lifestyle changes [2]. Cirrhosis led to about 1.5 million deaths in 2019 and accounts for 1.8% of the global incidence of disabilities per year [3,4]. Decompensated cirrhosis can also lead to other complications such as ascites, spontaneous bacterial peritonitis, esophageal variseal bleeding, hepatic encephalopathy, hepatorenal syndrome, and hepatocellular carcinoma [5].

Although the primary treatment for liver cirrhosis is managing the etiology and the complications of this condition, liver transplantation is still considered to be the best choice for select patients [6]. Indications for liver transplantation include cirrhosis in conjunction with decompensation. A Model for End-Stage Liver Disease (MELD) score of 15 or higher, or the presence of hepatopulmonary syndrome, are other indicators [7]. To address financial constraints and improve organ allocation, various models have been developed to assess the mortality risk of cirrhotic patients, such as the MELD [8]. The MELD is one of the most commonly used models for the prediction of the mortality of patients with cirrhosis, and new updates to it are being continually introduced [9].

The MELD scoring system utilizes serum creatinine and bilirubin and the International Normalized Ratio (INR) to assess the mortality risk of cirrhotic patients. However, recent studies have questioned its utility in the prediction of mortality in cirrhotic patients [10]. Further studies have shown that malnutrition is more severe in patients who expired within a shorter period of time [11]. Anthropometric parameters such as hand grip (HG), Mid-Arm Muscle Circumference (MAMC), and sarcopenia have been reported as indicators of malnutrition in cirrhotic patients [12].

Andrade et al. reported that diminished HG strength was related to a higher MELD score, demonstrating a correlation with a worse clinical outcome. The presence of decreased muscle strength in cirrhotic patients was illustrated to be linked to prognostic factors that can be valued as clinical data in the management of cirrhotic patients [13]. Ramachandran et al. evaluated the prevalence of malnutrition measuring HG and MAMC among cirrhotic patients referred for liver transplantation. They showed moderate-to-severe malnourishment, confirmed by low MAMC and HG measurements, which was associated with a MELD score of >15 and mortality [14]. Sarcopenia was shown to be associated with mortality in patients with cirrhosis. It is not correlated with the degree of liver dysfunction assessed by utilizing conventional scoring systems. Scoring systems including the assessment of sarcopenia are needed to better predict mortality in patients with cirrhosis [15]. In light of the previously mentioned evidence, and as anthropometric indices play important roles in predicting the prognosis of patients with cirrhosis [16,17], this study evaluated a new machine learning approach to predicting the three-month mortality probability in cirrhotic patients awaiting liver transplantation.

## 2. Materials and Methods

### 2.1. Devising a Prediction Scale in Cirrhosis

This prospective cohort study used a machine learning method that employed regularized Least Absolute Shrinkage and Selection Operator (LASSO) logistic regression to devise a prediction scale for three-month mortality in cirrhotic patients. The inclusion criteria for enrolled patients were as follows: being aged 18 years or older, having a Child–Turcotte–Pugh (CTP) score of 7 or higher, and having been referred to the Abu-Ali Sina Transplantation Center in Shiraz, Iran. The patient exclusion criteria were as follows: those who were younger than 18 years, those afflicted with HIV/AIDS, those suffering from Chronic Kidney Disease (CKD) and in need of Renal Replacement Therapy (RRT) such as hemodialysis, those afflicted with chronic pulmonary disease and in need of supplementary oxygen, pregnant and breastfeeding individuals, patients finding a liver donor and undergoing liver transplantation, and, finally, patients unwilling to participate in the study. All data were collected from the files of patients referred to Abu-Ali Sina Transplantation Center, Shiraz, Iran.

Since no previous study had employed our model in a similar context, the calculation of sample size was made based on a comparison of the Skeletal Muscle Mass Index (SMI) between surviving and deceased cirrhotic patients. Considering a type I error of 0.05 and a study power of 80% for male patients, the minimum sample size was estimated to be six participants. In this study, a census method was utilized and patients who needed a liver transplant during the one-year period were evaluated, which finally resulted in an adequate pool of patients. This study was approved by the Deputy of Research and Technology and the Committee of Ethics in Biomedical Research of Shiraz University of Medical Sciences, with the code IR.SUMS.REC.1400.220. After explaining the study to the patients and obtaining written consent, the eligible patients were enrolled in the study.

During physical examination, Mid-Arm Muscle Circumference (MAMC) was measured, and hand grip (HG) strength was calculated using a dynamometer. Average mean arterial pressure (AMAP) was determined using 24 h Holter monitoring. Moreover, the patients underwent spiral abdominopelvic CT scans, in which their SMI score was assessed. Since abdominopelvic CT scanning is routinely performed for cirrhotic patients, this procedure did not impose additional cost or radiation on the patients. Laboratory data such as serum creatinine, blood urea nitrogen (BUN), complete blood count (CBC), erythrocyte sedimentation rate (ESR), International Normalized Ratio (INR), serum sodium and potassium, and the complete liver function test (LFT) were evaluated. Then the patients were medically followed for three months.

### 2.2. The Inclusion and Exclusion Criteria

Patients aged 18 years or older with a Child–Turcotte–Pugh (CTP) score of 7 or higher who were referred to Abu-Ali Sina Transplantation Center were included in the study. The exclusion criteria were as follows: age under 18 years, affliction with HIV/AIDS, chronic kidney disease (CKD) requiring renal replacement therapy (RRT) such as hemodialysis, chronic pulmonary disease in need of supplementary oxygen, pregnancy, breastfeeding, and unwillingness to participate in the study. Patient recruitment occurred from December 2020 to September 2021, with medical follow-ups lasting until December 2021. The recruitment of the validation group of 21 patients lasted from January 2022 to April 2022, with follow-up extending to July 2022.

### 2.3. Statistical Analysis

The data were analyzed using the R statistical program (version 3.6.3, R Foundation for Statistical Computing). The quantitative variables were reported as means and standard deviations, while the qualitative data were reported as frequencies and percentages. Comparisons of quantitative variables between the deceased and surviving patient groups were conducted using independent two-sample t-tests, and categorical variables were compared using the chi-square test.

Variable selection was performed using regularized LASSO logistic regression, a machine learning method. Variables with non-zero coefficients (β) were included in the model, while variables with zero coefficients (β) were excluded (Table 1). Then, the maximum absolute beta value for each variable was calculated using the formula of $\left|\beta \right|=\beta \times Value\;range\;of\;the\;variable$ (Table 2).

The maximum absolute beta values were ranked first to kth (where *k* is the number of selected variables using the LASSO logistic regression model). The maximum absolute beta value of the top-ranked variable was used as the denominator for calculating each variable’s point contribution. The risk score equation was constructed by a linear interpolation method (Table 2). All statistical analyses were performed using the “rms” and “penalized” packages in R statistical software. The “rms” package of the R statistical program developed a graphic illustration for each variable’s point contribution and the overall risk score (Figure 1). Receiver Operating Characteristic (ROC) curves and their Area Under the Curve (AUC) values were used to compare the performance of the newly developed risk score with the MELD score (Figure 2). *p*-values of less than 0.05 were considered statistically significant.

This novel machine learning-based scoring system was named the Sina Score, in honor of the Persian physician Avicenna (Ibn Sina) who lived in the 10th and 11th centuries. To validate this new scoring system, 21 additional patients referred to our center were evaluated by both the Sina and MELD scores. ROC curves were generated for all participants to evaluate predictive accuracy.

## 3. Results

### 3.1. Demographic, Anthropometric, and Laboratory Findings

Among the 64 enrolled patients, 41 (64.1%) were male and 23 (35.9%) were female. The mean age of the patients was 46.50 ± 12.871 years. The patients were followed up for a period of three months, during which 13 (20.3%) died and 51 (79.7%) survived. The demographic, anthropometric, and laboratory findings for both groups are summarized in Table 1. As can be seen in Table 1, deceased patients were significantly older than survivors (53.77 ± 9.293 vs. 44.25 ± 13.252 years old, *p* = 0.006), had weaker hand grips (16.51 ± 8.682 vs. 24.14 ± 10.517 kg, *p* = 0.013) and had lower serum albumin and sodium levels (*p* = 0.049 and 0.022, respectively). The demographic, clinical, and laboratory results were assessed by univariate binary logistic regression analysis to predict mortality in cirrhotic patients. As can be seen in Table 1, older age (odds ratio [OR] = 1.073, *p* = 0.025), lower hand grip strength (OR = 0.913, *p* = 0.025), and lower sodium levels (OR = 0.874, *p* = 0.030) were significantly associated with increased risk of mortality. The results of the univariate analysis are summarized in Table 1.

### 3.2. The Prediction Scale in Cirrhosis

Since the sample size was low, regularized LASSO logistic regression was used to assess the key predictors of mortality in cirrhotic patients. Five variables turned out to have non-zero coefficients in LASSO regression analysis that entered our model, including HG, SMI, AMAP, Na, and total bilirubin (β column of Table 2). The risk of death was calculated using the formulae of linear predictors as follows:

Linear predictor = 19.8846 − 0.0956 × HG (kg) + 0.0351 × SMI − 0.0308 × AMAP (mmHg) − 0.1324 × Na (mEq/lit) − 0.0520 × Total bilirubin(mg/dL)$$Predicted\;risk\;of\;death=\frac{exp(Linear\;predictor)}{1+exp(Linear\;predictor)}$$

The results of this regression model were used to devise a new risk score by utilizing the maximal absolute coefficients of the variables. Table 2 summarizes the calculation of the points assigned to each variable in the risk score. Based on the findings in Table 2, the novel risk score was calculated using the following formula:

Risk Score = 624.69 − 2.0408 × HG + 0.7493 × SMI − 0.6575 × AMAP − 2.8263 × Sodium − 0.2772 × Total bilirubin.

### 3.3. Sina Scoring System

To increase the usability of the risk score, a nomogram for the Sina scoring system was generated by the R program (Figure 1) that helps estimate the risk of mortality on a visual scale. To compare the Sina score risk assessment tool with the MELD, the ROC and AUC were used and presented in Figure 2. The AUCs for Sina and MELD scores were 0.753 (95% Confidence Interval [95% CI]: 0.630–0.851) and 0.607 (95% CI: 0.478–0.725), respectively, showing the superior performance of the Sina score in predicting the risk of mortality in cirrhotic patients. At an optimal cutoff point of 195.6835, the Sina score could predict mortality with a sensitivity of 53.8% (95% CI: 25.2–80.7) and a specificity of 94.2% (95% CI: 84.0–98.7). The model had a 70.0% positive predictive value and an 89.1% negative predictive value. The high specificity of the Sina score has made it a potent risk stratifying tool for organ allocation in cirrhotic patients needing liver transplantation.

A linear interpolation method was used in the R statistical program to validate the Sina scoring system and draw a survival function graph. The graph drawn by the program showed a clear inverse relationship between the Sina score and three-month survival of cirrhotic patients. The linear interpolation graph is presented in Figure 3.

To assess the validity of the risk scoring system in the context of cirrhotic patients, a validation cohort of 21 cirrhotic patients was subsequently analyzed. The mean age of the validation group was 50.57 ± 12.221 years, and a male-to-female ratio of 5:2 was seen (71.4% of the patients were male, while 28.6% were female). No significant difference was seen between the development group and the validation group regarding age (*p* = 0.159) and sex (*p* = 0.485). Among the patients in the validation group, only two expired during a three-month interval. An ROC curve for the prediction of mortality using Sina and MELD scores was drawn for this population. The AUCs for MELD and Sina scores in the validation group were 0.706 (*p* = 0.173) and 0.825 (*p* = 0.032), respectively, demonstrating the superior predictive accuracy of the Sina scoring system. Figure 4 shows the ROC curve of the validation group.

## 4. Discussion

In this study, we evaluated a machine learning approach to predict mortality in patients with end-stage liver disease, aiming to improve organ allocation decisions in liver transplantation centers. The proposed Sina scoring system incorporates HG, SMI, AMAP during Holter monitoring, serum sodium, and total bilirubin levels.

Sarcopenia, defined as the loss of skeletal muscle mass, has been shown to be associated with chronic liver disease and an increase in the mortality of cirrhotic patients [18,19]. While esophageal variseal bleeding, hepatorenal syndrome, and hepatic encephalopathy are important predictors of acute mortality, sarcopenia reflects a more chronic deterioration in physiological reserve. Anthropometric indices such as body mass index (BMI), SMI, HG, and MAMC are utilized to assess sarcopenia and the nutritional status of the patient; thus, theses parameters were used in the current study to predict mortality in cirrhotic patients [20,21,22].

Furthermore, lower MAP was found to be independently associated with higher mortality in cirrhotic patients [23]. Consequently, we used the above-mentioned features to evaluate cirrhotic patients in our study. Serum sodium, total bilirubin and creatinine levels, and INR were also measured to evaluate the patients’ MELD scores as the current gold standard of prognostic evaluation in cirrhotic patients. Mallik et al. found that although the MELD is universally used for decision making in organ allocation for liver transplantation, the Chronic Liver Failure–Sequential Organ Failure Assessment (CLIF-SOFA) scoring system is superior in the prediction of short-term mortality among liver cirrhosis patients [24].

Our findings reveal that the MELD scoring system alone may impair medical judgment to determine the risk of mortality in cirrhotic patients. Although the MELD is globally used for the prediction of mortality in patients with cirrhosis, the addition of other variables in this scoring system can enhance its accuracy, such as adding C-Reactive Protein (CRP) and procalcitonin, which was demonstrated to improve the precision of mortality prediction in cirrhotic patients [25]. Similarly, Rodríguez-Perálvarez et al. found that the Gender-Equity Model for liver allocation corrected by serum sodium (GEMA-Na) and the Model for End-Stage Liver Disease 3.0 (MELD 3.0) could correct sex disparities for accessing liver transplantation and provide more accurate predictions of waiting list outcomes [26]. Also, the addition of lactate to the variables measured with the MELD score may make it superior in predicting mortality [27]. Deng et al. found that incorporating a new variable, such as five-meter gait speed as a measurement of frailty, in the MELD score could be associated with an increased predictive accuracy for mortality among cirrhotic patients [28]. The MELD scoring system incorporates serum creatinine levels to predict cirrhosis mortality, which is directly correlated with the muscle mass index of the patients; however, in our study, the SMI and HG were included for nutritional assessments of the patients, which could be the primary changes before creatinine alterations [29]. Moreover, Montano-Losza et al. performed a study on 669 cirrhotic patients and found out that the embodiment of sarcopenia assessed by the SMI at the level of third lumbar vertebra can increase the prognostic power of the MELD score for mortality [30].

In our study, malnutrition and frailty were measured by anthropometric indices, which were integrated into the Sina score. In a study performed by Tapper et al. on 274 patients with a 5-year follow-up, they found that body composition including, muscle and fat distribution, could outperform the MELD in predicting the risk of mortality and decompensation in cirrhotic patients. They found that fat density was the most important factor to predict decompensation and mortality in patients with end-stage liver disease [31]. These findings confirm our results regarding the importance of anthropometric indices in the assessment and prognosis of mortality in cirrhotic patients.

In another study performed by Kanwal et al., the authors developed new models for the prediction of mortality in cirrhotic patients by using LASSO regression containing numerous variables, such as hospitalization due to cirrhosis and the number of outpatient visits [32]. Our Sina score model, unlike the model used by Kanwal et al., consists of five core clinical parameters and provides a clear, usable formula for risk calculation, which increases its potential for bedside application. In a study by Maharshi et al., the authors found that the prevalence of malnutrition increased with higher CTP scores. Furthermore, the complications and mortality of cirrhosis in malnourished patients were more than that of well-nourished patients, revealing that nutritional assessments of cirrhotic patients can be of prognostic value in determining the risk of complications and mortality [33]. These findings denote the baseline concept that was used in the development of the Sina score as a new model for mortality risk assessment in cirrhotic patients.

Guo and colleagues conducted a study on a database of 34,575 cirrhotic patients utilizing multiple machine learning and deep learning methods to devise new prognostic models for mortality. These new models outperformed the conventional MELD scoring system in 3-month, 6-month, and one-year periods. However, since they did not provide a clear formula for the assessment of mortality risk, the clinical significance of the study is questionable, and using such data seems infeasible [34]. In contrast, our model retains interpretability crucial for clinical decision making. Finally, Chang et al. performed a meta-analysis on muscle mass loss and mortality in cirrhotic patients with a pooled population size of 4070 participants. They concluded that the loss of muscle mass had an adjusted hazard ratio of 2.36 in cirrhotic patients and an odds ratio of above 1.5 for advances in CTP class. Moreover, it was shown that patients with loss of muscle mass had a three-fold increase in infection risk [35]. As a result, the SMI as a measurement of muscle mass was demonstrated to be utilized as a means of mortality risk assessment.

## 5. Conclusions

In conclusion, we introduced and validated a new machine learning-based prognostic tool named the Sina score to predict three-month mortality in cirrhotic patients and help optimize transplant resources in decision making for liver allocation. By incorporating clinically relevant variables—including hand grip strength, skeletal muscle index, mean arterial pressure, serum sodium, and total bilirubin—this model demonstrated superior predictive accuracy compared to the traditional MELD score. These findings can improve risk stratification and patient selection for orthotopic liver transplantation.

## Figures and Tables

**Figure 1 jcm-14-04559-f001:**
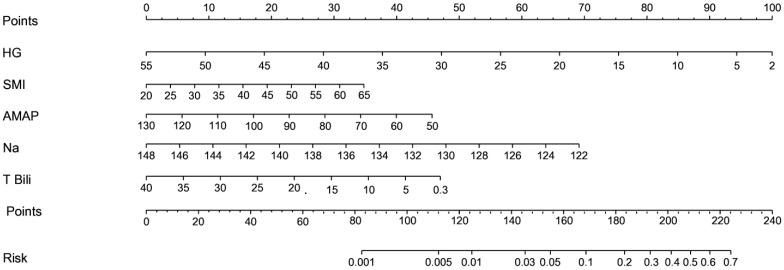
Nomogram of Sina score. AMAP: average mean arterial pressure; HG: hand grip; Na: sodium; SMI: Skeletal Muscle Mass Index; T. Bili: total bilirubin.

**Figure 2 jcm-14-04559-f002:**
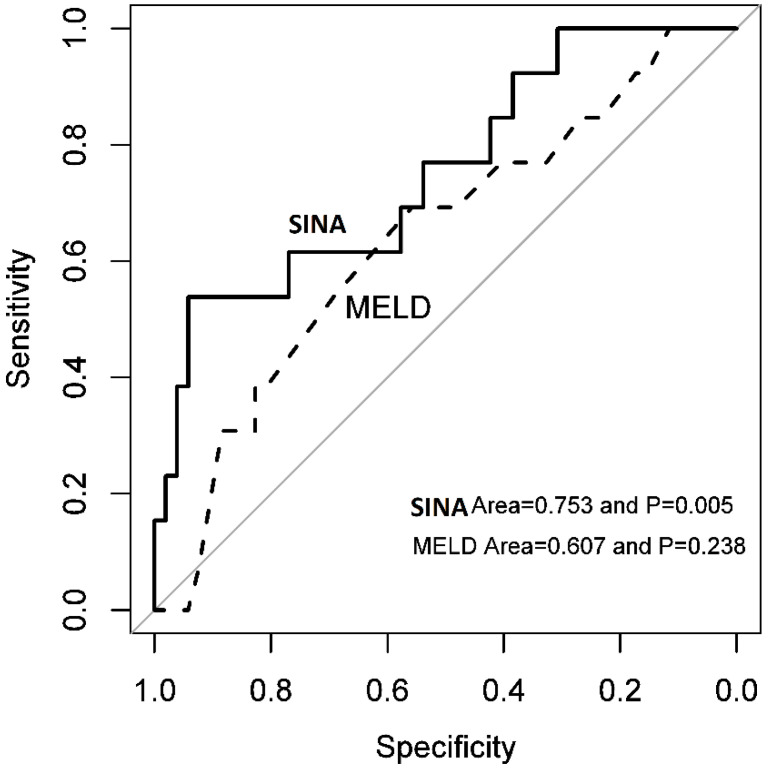
Comparison of the ROC for Sina and MELD scores.

**Figure 3 jcm-14-04559-f003:**
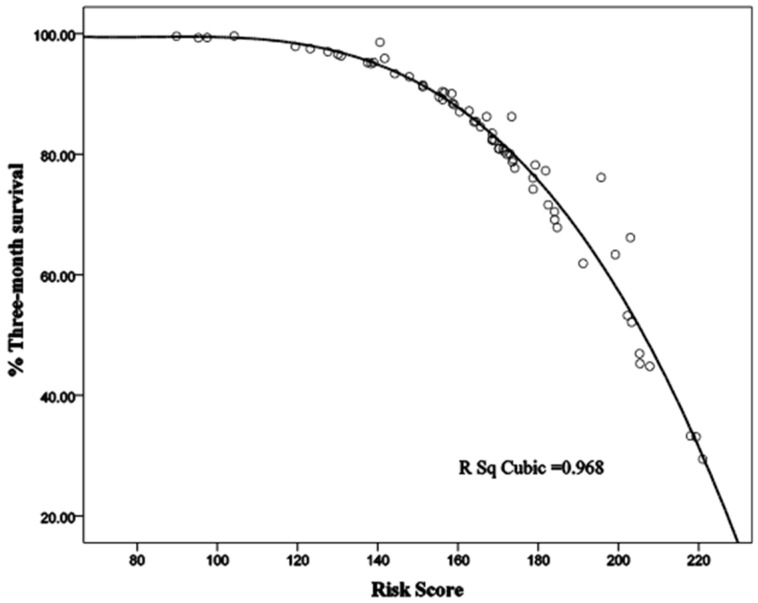
Estimated three-month survival rate as a function of Sina score.

**Figure 4 jcm-14-04559-f004:**
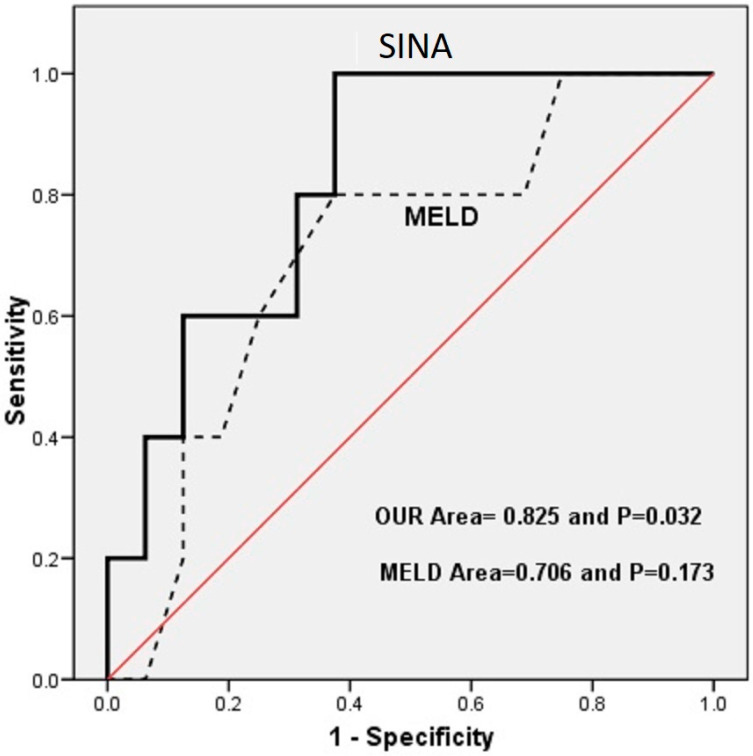
Comparison of the sensitivity and specificity of Sina and MELD scores in the validation group.

**Table 1 jcm-14-04559-t001:** Demographic and anthropometric findings and laboratory results of the enrolled patients.

Variable		Mean	SD	*p* Value	OR	*p* Value (ULR)
Age (years)	Expired	53.77	9.293	0.006	1.073	0.025
	Survived	44.25	13.252			
HG (kg)	Expired	16.51	8.682	0.013	0.913	0.025
	Survived	24.14	10.517			
MAMC	Expired	21.06	2.831	0.522	0.964	0.749
	Survived	21.33	2.725			
SMI	Expired	41.03	6.708	0.634	0.985	0.671
	Survived	42.16	9.120			
AMAP	Expired	76.85	16.582	0.093	0.940	0.093
	Survived	82.39	8.343			
WBC	Expired	6.38	2.288	0.475	0.061	0.576
	Survived	5.90	2.944			
Hemoglobin	Expired	11.49	1.706	0.313	0.858	0.314
	Survived	12.14	2.167			
Platelets	Expired	122.87	115.416	0.587	0.998	0.580
	Survived	142.65	115.865			
PTT	Expired	40.45	5.750	0.930	0.997	0.938
	Survived	40.62	7.438			
INR	Expired	1.62	0.462	0.764	0.881	0.806
	Survived	1.67	0.676			
BUN	Expired	20.77	10.305	0.170	1.024	0.267
	Survived	15.98	13.080			
Cr	Expired	0.90	0.234	0.756	0.659	0.755
	Survived	0.93	0.244			
Total protein	Expired	6.67	0.702	0.242	0.662	0.290
	Survived	6.94	0.867			
Albumin	Expired	3.09	0.461	0.049	0.477	0.130
	Survived	3.43	0.753			
AST	Expired	93.23	79.840	0.740	1.001	0.744
	Survived	84.77	85.633			
ALT	Expired	55.77	39.220	0.239	0.995	0.495
	Survived	79.40	120.052			
Total bilirubin	Expired	4.19	2.994	0.628	0.984	0.756
	Survived	4.83	7.272			
Direct bilirubin	Expired	1.86	2.195	0.490	0.956	0.630
	Survived	2.46	4.301			
Alkaline Phosphatase	Expired	382.52	227.033	0.906	1.000	0.927
	Survived	392.18	367.072			
Na	Expired	135.25	4.996	0.022	0.874	0.030
	Survived	138.83	4.893			
K	Expired	4.24	0.232	0.993	0.994	0.993
	Survived	4.24	0.585			

ALT: Alanine Aminotransferase; AMAP: average mean arterial pressure; AST: aspartate aminotransferase; BUN: blood urea nitrogen; Cr: creatinine; HG: hand grip; INR: International Normalized Ratio; K: potassium; MAMC: Mid-Arm Muscle Circumference; Na: sodium; OR = odds ratio; PTT: partial thromboplastin time; SD: standard deviation; SMI: Skeletal Muscle Mass Index; ULR: univariate logistic regression; WBC: white blood cell.

**Table 2 jcm-14-04559-t002:** Estimated LASSO regression coefficient and assigned points for each feature.

Variable	β	Range of Variable	$\left|\beta \right|$ *	Rank	Assigned Point	Linear Interpolation Equation
HG	−0.0956	51 to 2 by 5	4.6844	1	0 assigned to HG = 51 100 × (4.6844/4.6844) = 100 assigned to HG = 2	Point = 104.08 − 2.0408 × HG
SMI	0.0351	22 to 63 by 5	1.4391	5	0 assigned to SMI = 22 100 × (1.4391/4.6844) = 30.72 assigned to SMI = 63	Point = −16.48 + 0.7493 × SMI
Average MAP	−0.0308	123 to 50 by 10	2.2484	3	0 assigned to AMAP = 123 100 × (2.2484/4.6844) = 48.00 assigned to AMAP = 50	Point = 80.88 − 0.6575 × AMAP
Na	−0.1324	147 to 123 by 2	3.1776	2	0 assigned to Na = 147 100 × (3.1776/4.6844) = 67.83 assigned to Na = 123	Point = 415.46 − 2.8263 × Sodium
Total bilirubin	−0.0520	37.02 to 0.42 by 5	1.9032	4	0 assigned to total bilirubin = 147 100 × (1.9032/4.6844) = 40.63 assigned to total bilirubin = 0.42	Point = 40.75 − 0.2772 × Total bilirubin

β: estimated regression coefficient; $\left|\beta \right|$: absolute maximum beta value. Abbreviations: AMAP: average mean arterial pressure; HG: hand grip; LASSO: Least Absolute Shrinkage and Selection Operator; * $\left|\beta \right|$ = β × value range of the variable. MAMC: Mid-Arm Muscle Circumference; Na: sodium; SMI: Skeletal Muscle Mass Index.

## Data Availability

The datasets analyzed during the current study are available from the corresponding author by e-mail at sjm@sums.ac.ir on reasonable request.

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
