# Peer review of "Sina Score as a New Machine Learning-Derived Online Prediction Model of Mortality for Cirrhotic Patients Awaiting Liver Transplantation: A Prospective Cohort Study"

_jcm, 2025, doi:10.3390/jcm14134559_

Round 1
Reviewer 1 Report
Comments and Suggestions for Authors
It's interesting to develop new models besides MELD to predict survival in cirrhosis patients. The authors used LASSO regression to establish a new model-defined Sina scoring system and compared its efficiency with MELD in predicting mortality in patients with end-stage liver disease.
Nevertheless, several important points should be noted.
1. Regarding the patients included in the "Materials and Methods" section, it's unclear which part of the cirrhotic population was included in this study. Whether these patients had received LT/no LT?
2. If patients with cirrhosis after LT were included in this study, it's not clear at what point in time the authors collected the clinical characteristics of the patients.
3. How did the authors calculate the size of the sample used in this study? The sample size in this study was small to perform and validate the predicted model, which reduces the power of results.
4. There was a sexual dimorphism in the size of men and women. Whether there was a sex bias in the predicted efficacy of the Sino score including MAMC, HG and SMI.
5. It's important to evaluate the specificity of the Sina score for cirrhotic patients, especially for those undergoing LT. The authors should make a comparison with the group of cirrhotic patients without LT.
6. Major review.
Author Response
Dear Mr. Avery Su
jcm@mdpi.com
avery.su@mdpi.com
MDPI Branch Office, Haidian, Beijing
Floor 13, Gengfang International Building, No.13 Huayuan Road, Haidian
District, 100088 Beijing, China
Tel. +86 010 62563691
Hello
Thanks for your kind follow up regarding our submission as follows:
Manuscript ID: jcm-3383665
Type of manuscript: Article
Title: Sina Score as a New Machine-Learning-Derived Online Prediction Model
of Mortality for Cirrhotic Patients Undergoing Liver Transplantation: A
Retrospective Cohort Study
Authors: Seyed Mohammad Kazem Hosseini-Asl, Seyed Jalil Masoumi, Ghazaleh
Rashidizadeh, Amir Hossein Hassani, Golnoush Mehrabani *, Vahid Ebrahimi,
Seyed Ali Malek-Hosseini, Saman Nikeghbalian, Alireza Shakibafard
Received: 5 Dec 2024
- I wish to let you know that changes were included and yellow highlighted in the text point to point.
- Detailed Author Contribution Form was also provided.
- Respective ethical committees approvals of the study (Ethical code: IR.SUMS.REC.1400.220) has been mentioned in the text.
- All references were checked and verified to be relevant to the content of the manuscript.
- English structure was rechecked to ensure the spelling and grammar.
Review Report Form
Open Review
(x) I would not like to sign my review report
( ) I would like to sign my review report
Quality of English Language
(x) The English is fine and does not require any improvement.
( ) The English could be improved to more clearly express the research.
Yes |
Can be improved |
Must be improved |
Not applicable |
|
Does the introduction provide sufficient background and include all relevant references? |
( ) |
(x) |
( ) |
( ) |
Is the research design appropriate? |
( ) |
( ) |
(x) |
( ) |
Are the methods adequately described? |
( ) |
( ) |
(x) |
( ) |
Are the results clearly presented? |
( ) |
( ) |
(x) |
( ) |
Are the conclusions supported by the results? |
(x) |
( ) |
( ) |
( ) |
Comments and Suggestions for Authors
It's interesting to develop new models besides MELD to predict survival in cirrhosis patients. The authors used LASSO regression to establish a new model-defined Sina scoring system and compared its efficiency with MELD in predicting mortality in patients with end-stage liver disease.
Nevertheless, several important points should be noted.
- Regarding the patients included in the "Materials and Methods" section, it's unclear which part of the cirrhotic population was included in this study. Whether these patients had received LT/no LT?
Patients who underwent liver transplantation were excluded since it was not clear whether they could survive for three months or not. This study was performed just on the population of patients who NEEDED to undergo liver transplantation but were not successful to receive a viable liver transplant.
- If patients with cirrhosis after LT were included in this study, it's not clear at what point in time the authors collected the clinical characteristics of the patients.
Patients who underwent liver transplantation were excluded from the study.
- How did the authors calculate the size of the sample used in this study? The sample size in this study was small to perform and validate the predicted model, which reduces the power of results.
No previous study used the mentioned mathematical model but for comparison of SMI between expired and survived cirrhotic patients; only six patients would be sufficient. However, we used the data of all patients referred to our referral center during one-year period. Of course, for better validation of the efficacy of this prediction model; the data of other clinical centers and international collaborations seem necessary.
- There was a sexual dimorphism in the size of men and women. Whether there was a sex bias in the predicted efficacy of the Sino score including MAMC, HG and SMI.
It is true that the hazard ratios for mortality using SMI should be adjusted based on sex; however, due to the limited sample size, adjustment of this variable would distort the findings. Furthermore, for better judgement of the efficacy of SINA score in men and women; further studies with larger sample sizes preferable from various ethnic origins are essential
- It's important to evaluate the specificity of the Sina score for cirrhotic patients, especially for those undergoing LT. The authors should make a comparison with the group of cirrhotic patients without LT.
Patients who underwent liver transplantation were excluded from the study.
- Major review.
Sincerely Yours
Golnoush Mehrabani, MD, PhD
Reviewer 2 Report
Comments and Suggestions for Authors
This study explores the use of a machine-learning-derived score, the Sina Score, to predict three-month mortality in cirrhotic patients undergoing liver transplantation. The researchers compared the performance of the Sina Score to the MELD score using a retrospective cohort of 64 patients, finding it to be an equally precise prognostic tool. Key variables included hand grip strength, skeletal muscle mass index, mean arterial pressure, serum sodium, and total bilirubin.
This is an intriguing study addressing an important topic—the need for improved prediction models for cirrhotic patients undergoing liver transplantation. However, the small sample size of 64 patients is a significant limitation and raises questions about the generalizability of the findings.
Clarification of Sina Score: The Sina Score is not adequately introduced or explained in the introduction or methods sections. Readers need a clearer understanding of its derivation, clinical rationale, and how the machine learning model was applied. Was it derived entirely from the LASSO regression? A detailed explanation is crucial for reproducibility and credibility.
Literature Context and Methodology Standards: The authors should consider reviewing additional literature to contextualize their findings better, especially existing predictive models in liver transplantation and cirrhosis. Following the TRIPOD (Transparent Reporting of a multivariable prediction model for Individual Prognosis Or Diagnosis) statement could significantly improve the manuscript’s quality. This statement outlines best practices for reporting prediction model studies, ensuring transparency and completeness.
Advanced Therapy Modalities: It would be helpful to compare the Sina Score's performance to other advanced predictive tools or machine learning algorithms in the field. Expanding the discussion to include this could add depth and context to the study: https://pmc.ncbi.nlm.nih.gov/articles/PMC10941741/
Typos and Presentation: The manuscript contains several minor errors, including awkward phrasing and unclear abbreviations. For instance, the phrase "So Sina score can be recommended as precise as MELD" is not grammatically correct and should be revised.
Comments on the Quality of English Language
Additionally, the language throughout the manuscript appears amateurish and could benefit from substantial editing for clarity and professionalism.
Author Response

(The authors gave the same response as above.)

Reviewer 3 Report
Comments and Suggestions for Authors
This article was difficult to follow and read because it wasn't uploaded according to MDPI standards. It didn't have numbered lines, but I tried to make a review as pertinent as possible.
Introduction
1. "The prevalence and incidence of cirrhosis has an increasing trend due to lifestyle changes [2]."
=> Use active voice for clarity: "The prevalence and incidence of cirrhosis are increasing due to lifestyle changes." Please briefly mention specific lifestyle changes like obesity, alcohol consumption.
2. "Cirrhosis has led to about 1.5 million deaths in 2019 and accounts for 1.8% of the global disabilities per year [3,4]."
=> Consider rephrasing for precision: "In 2019, cirrhosis caused approximately 1.5 million deaths and accounted for 1.8% of global disabilities annually."
3. "Decompensated cirrhosis can also lead to other complications such as ascites, spontaneous bacterial peritonitis, esophageal varicose bleeding, hepatic encephalopathy, hepatorenal syndrome, and hepatocellular carcinoma [5]."
=> Avoid long lists unless necessary. Group related complications for conciseness.
4. "Although the primary treatment for liver cirrhosis is managing the etiology and the complications of this condition, liver transplantation is still considered as the best choice for selected patients [6]."
=> Simplify: "Managing the underlying cause and complications is the primary treatment for liver cirrhosis, but liver transplantation remains the best option for selected patients."
5. "Indications for liver transplantation include cirrhosis in conjunction with decompensation. A Model for End-stage Liver Disease (MELD) score of 15 or higher, or suffering from hepatopulmonary syndrome can be mentioned among other indicators [7]."
=> Improve sentence clarity by restructuring: "Indications for liver transplantation include decompensated cirrhosis, a MELD score of 15 or higher, or the presence of hepatopulmonary syndrome."
6. "To address financial constraints and improve organ allocation, various models have been developed to assess the mortality risk of cirrhotic patients such as MELD [8]."
=> Specify how MELD addresses these issues to provide more context.
7. "However, recent studies have questioned its utility in prediction of mortality in cirrhotic patients [10]."
=> Specify why its utility is questioned—this could strengthen the argument.
8. "Further studies have shown that malnutrition is more severe in patients who were expired during a shorter period of time [11]."
=> Avoid "expired" for deceased patients, as it may seem informal or insensitive. Suggest: "who passed away within a shorter period."
Materials and Methods
1. "This retrospective cohort study used a machine learning method by employing the regularized Least Absolute Shrinkage and Selection Operator (LASSO) logistic regression to devise a prediction scale for three-month mortality in cirrhotic patients."
=> The sentence is clear, but it could benefit from briefly explaining why LASSO was chosen over other methods. For example: "LASSO was selected due to its efficiency in variable selection and predictive accuracy."
2. "All data were collected from file of patients referred to Abu-Ali Sina Transplantation Center, Shiraz, Iran."
=> Correct grammatical inconsistency: "All data were collected from the files of patients referred to the Abu-Ali Sina Transplantation Center in Shiraz, Iran."
Inclusion and Exclusion Criteria
3. "The inclusion criteria were being patients aged 18 years and older with Child-Turcotte-Pugh (CTP) scores of 7 or higher who referred to Abu-Ali Sina Transplantation Center."
=> Consider rephrasing for conciseness: "Patients aged 18 years or older with CTP scores ≥7 who were referred to the Abu-Ali Sina Transplantation Center were included."
4. "The exclusion criteria were patients younger than 18 years old, affliction with HIV/AIDS..."
=> Rephrase for consistency: "Exclusion criteria included patients under 18 years old, those with HIV/AIDS, chronic kidney disease requiring renal replacement therapy, or chronic pulmonary disease requiring supplemental oxygen."
Statistical Analysis
5. "The data were entered into R Statistical Program (Version 3.6.3, R Foundation for Statistical Computing)."
=> Specify any additional software packages or libraries used (e.g., "rms" and "penalized" packages mentioned later).
6. "Variable selection was achieved using regularized LASSO logistic regression as a machine learning method."
=> Consider explaining what non-zero coefficients represent in LASSO for readers unfamiliar with the method.
Results
Demographic, Anthropometric, and Laboratory Findings
1. "Among 64 enrolled patients, 41 (64.1%) were male and 23 (35.9%) were female."
=> Add interpretation: Does this male-to-female ratio reflect known trends in cirrhosis prevalence? The Prediction Scale in Cirrhosis
2. "To make a prediction model for mortality in cirrhotic patients..." ]
=> Consider specifying why the five selected variables were included in the final model (HG,
Author Response

(The authors gave the same response as above.)

Reviewer 4 Report
Comments and Suggestions for Authors
This retrospective study evaluated the efficacy of the machine learning developed Sina score (Na, bil, HGS, SMI, MAP) in the mortality prediction of patients with liver cirrhosis on the transplant list. The authors compared it with the MELD score and found even better performance in the prediction of the 3 month mortality. The study is well designed, clinically important and interesting. The references are fine. The tables are in order. There are 3 points to consider:
1. The measurement of SMI, hand grip and MAMC should be described in more detail in the methodology section.
2. Please discuss the MELD-sarcopenia score in more detail and compare it with the Sina score.
3. This score is more time consuming and more expensive than the MELD score measurement, therefore it will be harder to implement in the clinical practice, especially regarding the follow up of these patients- please comment.
Author Response

(The authors gave the same response as above.)

Round 2
Reviewer 2 Report
Comments and Suggestions for Authors
The authors did not fully anwer my queries.
Comments on the Quality of English LanguageNeeds improvement.
Reviewer 3 Report
Comments and Suggestions for Authors
I'm ok with the final form of the article